# Salt Reduction Intervention in Families Investigating Metabolic, Behavioral and Health Effects of Targeted Intake Reductions: Study Protocol for a Four Months Three-Armed, Randomized, Controlled “Real-Life” Trial

**DOI:** 10.3390/ijerph16193532

**Published:** 2019-09-21

**Authors:** Kirsten Schroll Bjoernsbo, Nanna Louise Riis, Anne Helms Andreasen, Janne Petersen, Anne Dahl Lassen, Ellen Trolle, Amalie Kruse Sigersted Frederiksen, Jens Kristian Munk, Ulla Toft

**Affiliations:** 1Center for Clinical Research and Prevention, Bispebjerg and Frederiksberg Hospital, 2000 Frederiksberg, Denmark; Nanna.Louise.Riis@regionh.dk (N.L.R.); Anne.Helms.Andreasen@regionh.dk (A.H.A.); Janne.Petersen.01@regionh.dk (J.P.); amalieksf@gmail.com (A.K.S.F.); Ulla.Toft@regionh.dk (U.T.); 2Division of Risk Assessment and Nutrition, National Food Institute, Technical University of Denmark, 2800 Kgs. Lyngby, Denmark; adla@food.dtu.dk (A.D.L.); eltr@food.dtu.dk (E.T.); 3Section of Biostatistics, Department of Public Health, University of Copenhagen; Øster Farimagsgade 5 Entrance B, 2nd floor, 1014 Copenhagen K, Denmark; 4Department of Clinical Biochemistry, Amager and Hvidovre Hospital, 2650 Hvidovre, Denmark; Jens.Kristian.Munk@regionh.dk

**Keywords:** gradual salt reduction, sodium potassium ratio, cardiovascular consequences, randomized controlled trial, statistical analysis plan

## Abstract

Reductions in salt intake have the potential to markedly improve population health at low cost. Real life interventions that explore the feasibility and health effects of a gradual salt reduction lasting at least four weeks are required. The randomized controlled SalT Reduction InterVEntion (STRIVE) trial was developed to investigate the metabolic, behavioral and health effects of four months of consuming gradually salt reduced bread alone or in combination with dietary counselling. This paper describes the rationale and methods of STRIVE. Aiming at 120 healthy families, participants were recruited in February 2018 from the Danish Capital Region and randomly allocated into: (A) Salt reduced bread; (B) Salt reduced bread and dietary counseling; (C) Standard bread. Participants were examined before the intervention and at four months follow-up. Primary outcome is change in salt intake measured by 24 h urine. Secondary outcomes are change in urine measures of potassium and sodium/ potassium ratio, blood pressure, plasma lipids, the renin-angiotensin system, the sympathetic nervous response, dietary intake as well as salt taste sensitivity and preferences. The results will qualify mechanisms affected during a gradual reduction in salt intake in compliance with the current public health recommendations.

## 1. Introduction

Excess dietary sodium has a major role in the pathogenesis of hypertension, a leading risk factor for premature death in the world [1,2]. The dietary salt intake in most countries worldwide, including Denmark, is far beyond the recommended level (<5–6 g/day) [3] and population-based salt reduction has been rated to be one of the most cost-effective strategies to prevent cardiovascular disease (CVD) [4,5]. A lower salt intake, if continued, may lessen the subsequent rise in blood pressure (BP) with age, which could have major public health implications in terms of preventing the development of hypertension and CVD later in life [6]. Therefore, a 30% reduction in mean population intake of salt is among the 10 global targets in the Global Action Plan 2013–2020 by the World Health Organization to prevent non-communicable diseases [3]. Hence, in a report developed for the Danish Ministry of Food, Agriculture and Fisheries, we have estimated that a reduction in the mean salt intake of 3 g/day in the Danish population would reduce fatal and non-fatal new cases of CVD by 6–10% annually, which in the long term means reduced health care costs by up to 296 million USD per year [7].

Fewer studies in this area are done among children. However, excess salt intake during early life leads to taste preferences for saltier foods [8] and there is good evidence that salt intake plays an important role in regulating BP in children too [9]. Although the magnitude of the association between salt reduction and BP in children is relatively modest, BP follows a tracking pattern from childhood into adulthood; therefore, the public health benefits of shifting the distribution of population levels are important [10].

Several countries including Denmark have initiated national programs to decrease salt intake in the population [11,12]. However, some studies have shown a U-shaped association between sodium intake and CVD and currently the effect of salt reduction on morbidity and mortality in the population is heavily discussed [11,13,14,15]. The findings of potential adverse effects of salt reduction on the risk of CVD have been argued to be due to e.g., suboptimal measurement of salt intake and bias attributable to indication or reverse causality [16]. Still, some randomized studies have shown that when intake is reduced, there is a physiological stimulation of the renin-angiotensin-aldosterone system and the sympathetic nervous system, which may again increase risk of CVD [17]. However, these findings are mainly based on very short-term studies with a very large acute salt reduction and thus there is a need for studies that explore these mechanisms during a modest reduction in salt intake for a longer period analogously to the current public health recommendation of gradual salt reduction [1].

Potassium has the potential to counteract some of the negative consequences of a high sodium intake on especially BP [18,19,20]. Some earlier studies have found that a high sodium/potassium ratio is a stronger risk factor for cardiovascular disease than levels of either sodium or potassium alone. The role of potassium is not yet well understood [21].

In addition to a better understanding of the health effects of a reduced salt intake, there is a need for better evidence regarding effective strategies to reduce salt intake in Denmark and other countries. National programs to reduce salt intake on a population-based level relies mainly on food product reformulation as up to 80% of the salt intake in western countries comes from salt added to processed foods [22]. Like in the USA and other western countries the biggest contributor of dietary sodium intake in Denmark is bread [23]. Therefore, reducing the sodium content in bread could be a relevant part of an effective strategy to reduce the sodium intake of the population. Because sudden large reduction of salt can make foods unacceptable to consumers [24], a gradual reduction of the salt content is generally recommended [3].

The reformulation intervention strategy makes it possible to investigate the effect of salt reduction exclusively in a single-blinded design, without changing other dietary factors. Thus, the conclusions on salt reduction will be strong and transferable to real life. Furthermore, targeting families rather than individuals improves implementation, as the setting reflects the life situation that salt is consumed in, which is essential to achieve dietary goals [25].

To meet the national goal of lowering salt intake by 3 g/d, reformulation of a single food cannot stand alone. Dietary counseling is another possible strategy to reduce salt intake. It is a relevant strategy to influence knowledge, attitude and behavior related to salt, while simultaneously influencing potassium intake. A combined intervention strategy is potentially the most efficient strategy to reach the recommended daily level of salt intake.

To our knowledge, no previous trials have investigated the impact of a gradual salt reduction in healthy participants. Thus, we designed a real-life trial to test whether salt reduction of bread alone or in combination with dietary counselling are effective strategies to reduce salt intake in families as measured by 24 h urine collection. To elucidate any beneficial as well as adverse effects of a gradual salt reduction, the metabolic, behavioral and health consequences will be compared between families receiving different salt reduction strategies and families receiving standard bread. More specifically the objectives of the intervention are:To examine the effects of two different salt reduction strategies ((A) gradually lowering salt content in bread; (B) intervention combining salt reduced bread and dietary counseling) on intake of salt, sodium and potassium intake, the sodium/potassium ratio and the overall dietary intake,To examine the effects of different strategies of salt reduction on selected cardiovascular risk factors (blood pressure, blood lipids, renin, aldosterone, norepinephrine, epinephrine),To test effects of different levels of salt reduction on salt sensitivity and preference.

The aim of this paper is to describe the rationale, the methods and the statistical analysis plan of this trial, so that this information is made public.

## 2. Materials and Methods

### 2.1. Study Design

This study, called SalT Reduction InterVEntion (STRIVE) trial is a single blinded, cluster randomized controlled trial, with children and adults recruited as families and randomly assigned to three parallel intervention arms. Outcome measures were assessed at baseline (before bread intervention start) and four months later, while still consuming intervention bread, hereafter the intervention ended. Primary outcome measures (salt intake and biomarkers) was collected by blinded assessors. The trial flow is shown in Figure 1 and an overview of the trial characteristics is provided in Appendix A.

### 2.2. Study Setting, Sample and Recruitment Procedure

In January–March 2018 families were recruited through digital and paper advertisements posted in social media, schools, kindergartens and companies sited in five municipalities (Albertslund, Ballerup, Egedal, Glostrup and Rødovre) in the southwestern part of the Capital Region of Denmark. The area was selected to be close to either the local bakery that provided the bread, or Center for Clinical Research and Prevention (CCRP), Glostrup Hospital, where the health examinations took place.

Interested families were screened for eligibility and invited for an introduction meeting, where they received a thorough verbal information about the trial. Written information was sent to families before the meeting.

Inclusion criteria: Families of minimum one child (aged 3–17 years) and one parent (aged 18–69 years), living together and with a daily intake of bread (minimum 175 g of bread per day in adults). Adult siblings living at home, could also join the study—if the first criterion was met. Children (aged 3–17 years), living fifty percent of their time with each of their divorced parents (aged 18–69 years), could participate if both parents joined the study.

Exclusion criteria: Antihypertensive treatment, lipid-lowering treatment, cardiovascular disease, diabetes, pregnancy, urine albumin measured in spot urine at baseline greater than 300 mg/day, celiac disease or gluten intolerance.

### 2.3. Ethics

Informed consent was obtained from all primary caregivers and adults before participation in the study. Before signing the consent forms, participants were informed that participation in the trial was voluntary and they could withdraw anytime. The study was conducted in accordance with the Declaration of Helsinki and the protocol was approved by The Scientific Ethical Committee, Capital Region, Denmark (project indication code H-17030995 on 27th December 2017). STRIVE was approved by the Danish Data Protection Agency (CSU-FCFS-2017-017) and registered at ClinicalTrials.gov (NCT03810885).

### 2.4. Randomization

Once included, baseline assessments were conducted before randomization of the families into one of three groups (A: salt reduced bread, B: salt reduced bread and dietary counseling, C: standard bread no dietary counseling) by a ratio of 1:1:1. Randomization followed a computer-generated sequence of random group assignments (allocation sequence) produced by the data manager. In divorced families, both parents (and their families) followed the randomization of the parent first examined.

### 2.5. Blinding

Participants were blinded to the treatment they received. Bread was coded by a red or blue label, and families collected bread according to the color of their group. Color-code for the bread was revealed after the last participant completed the follow-up measurements.

The color code of the bread was known by two bakers responsible for baking the bread but blinded to the rest of the staff in the bakery, including those handing out the bread to the participants.

The secretary screening and enrolling the participants was blinded with respect to group allocation. The three research nurses conducting the clinical measurements, at baseline (before randomization) and at follow-up remained blinded to the group allocation. Salt sensitivity and preference were tested before randomization at baseline, and therefore unknown to the assessors, but one assessor knew the group allocation, at follow up. Given the nature of the intervention it was not possible to blind the nutritionists delivering the dietary counselling. However, the primary outcome measures were blinded to the outcome assessors. All data was entered with error control in the application or using double entry.

### 2.6. Sample Size

The average daily intake of bread in Denmark is 138 g for adults 18–75 years of age and 128 g for children 4–17 years of age [26]. One of the inclusion criteria was a daily intake of bread of minimum 175 g among adults, corresponding to P75 intake among adults. The average salt content of bread in Denmark was 1.3 g/100 g of bread [27]. Thus, the average daily intake of salt from bread among the adult participants was estimated to be approximately 2.3 g. The expected reduction in salt intake due to the salt-reduced-bread intervention (average reduction of 55%) was 1.25 g per day. The expected salt reduction in the intervention group with the combined intervention was on average 3 g (based on the current average daily salt intake and the target for the intervention). The statistical power calculations were built on data from The Danish baseline study of Functional Disorders (DanFunD) showing an average daily intake of salt of 8.3 g (SD 2.2) [28]. The analyses were done to fit the cluster-randomized design. A family was assumed to consist of four members and the intra-class correlation (ICC) within families was assumed to be 0.33. With a difference at follow-up of 1.2 g of salt per day between (salt-reduced) intervention and control group we needed to include 25 families (100 persons) in each group to be able to reject the null hypothesis at a 5% level with a power of 0.8. With an expected difference in salt intake of 1.2 g/day we expected to see clinically relevant differences in blood pressure, blood lipids, renin, aldosterone and albuminuria.

### 2.7. Study Principles

This protocol was reported in accordance with the Standard Protocol Items: Recommendations for Interventional Trials (SPIRIT) 2013 Statement [29]. The reporting of the study will follow the Consolidated Standards of Reporting Trials (CONSORT) statement using the extension for non-pharmacological trials [30].

### 2.8. Intervention Overview

Bread was delivered during an approximately four months period. Twice a week each family received a rye bread, wheat bread and ten wheaten buns. Over the first six weeks salt content was reduced by 0.2 g/100 g until a minimum of 0.6 g/100g in rye bread and 0.4 g/100g in wheat bread in trial A (see Table 1).

The first two weeks all bread was kept at the standard level, until all families were ready to enter the intervention. Participants was instructed to replace their usual consumption of bread by bread products provided in the study. In all other respects, participants were requested to live their lives as usual, however participants receiving dietary counseling had to adjust their diet according to the advice given. Families received the intervention bread for free and a ticket to the cinema after completion of the intervention.

#### 2.8.1. Intervention A: Salt Reduced Bread

After the first two weeks (run-in) on regular bread participants in the salt-reduced bread group were provided with bread that was gradually reduced in salt content the following four weeks. The intervention continued with a 67% reduction in wheat bread and 50% reduction in rye bread for approximately three months.

#### 2.8.2. Intervention B: Salt Reduced Bread Combined with Dietary Counseling 

In addition to salt reduced bread, participants in this group received: (1) a group introduction on how to reduce salt and increase potassium intake followed by (2) an individual goal setting meeting in the family, followed by two follow-up phone calls and (3) a weekly newsletter reinforcing salt reduction:

(1) The group introduction started with a 30 min verbal version of the national guidelines on salt and hypertension. Participants filled out a salt-screener (developed for this study) and joined workshops on nutrition declaration and food labelling followed by a flavor workshop and a low salt tapas bar.

(2) Families were invited to an individual dietary counseling meeting at the CCRP or at home. Based on the salt screener the nutritionist identified foods or meal patterns, where sodium could be reduced, or potassium increased. At the meeting these patterns were further discussed with the families, whereby the families developed focus points (goals) for their salt reduction and evaluated how motivated they were for reducing dietary salt. After 1–2 weeks the nutritionist called the families to follow up on the progression and reinforced the salt reduction. Barriers were identified and ways to get around them discussed. New focus points were agreed upon if needed. A final follow-up call was performed after another 1–2 weeks.

(3) Weekly emails with educational material on how to reduce salt intake were send to the families. Information given at introduction or individual meetings was further elaborated in the emails.

#### 2.8.3. Control Group

The control group received bread with a regular content of salt and no dietary counseling.

### 2.9. Bread Production

Bread was developed in cooperation with Lantmännen Cerealia and a local bakery. Lantmännen had experience from a former study with fortified bread [31] and developed recipes based on existing breads from the local bakery. Before intervention start, the bakery gradually reduced the salt content of their rye bread from about 1.7 g/100 g to meet the keyhole label maximum salt content of 1.2 g/ 100 g. The keyhole label is a Nordic label helping consumers to choose healthier foods. Foods labelled with the Keyhole symbol contain less fat, sugars and salt and more dietary fibre than food products of the same type not carrying the symbol [32]. It was not possible to reduce the salt content in rye bread below 0.6 g/100 g without losing the usual texture. To develop as much taste in the bread as possible, sour dough, long rising time and seeds were applied. As salt was reduced, traditional salt was replaced by sea salt, added as a liquid solution. This was done to increase salt taste slightly even at low concentrations of salt. Bread was baked and packed in one production line, to ensure that different salt contents were not mixed up. Samples of each bread type were analyzed for salt content every week by the Danish Veterinary and Food Administration Inspection Laboratory.

Families were able to adjust the amount of bread delivered during the intervention and choose the type of rye bread they preferred. For the sake of variation different types of wheat bread were baked over the weeks. Each week, both the rye bread and one of the wheat recipes fulfilled the criteria of the whole grain partnership [33].

### 2.10. Theoretical Framework of the Dietary Counseling Intervention

A dietary counseling program was developed as an addition to the salt reduced bread, in accordance to further reducing dietary salt and increasing potassium through counselling. It was given by six trained MSc’s with a nutrition background (see Table 2). For salt-reduction-advice-programs to be successful, consumers need clear messages on the rationale behind changing sodium and potassium intake. Also, practical and culturally appropriate means to identify products with lower salt content and skills to cook meals with less salt but still rich in flavor are needed [34]. To achieve this five key messages were recurring through the counselling: (1) Buy less salty foods by checking nutrition facts labels or go for the keyhole label; (2) Eat less of foods high in salt like meat cuts, cheese, convenience foods; (3) Flavor foods without salt with herbs and spices; (4) Add less salt during cooking and avoid adding salt at the table; (5) Follow the plate model for fruit and many vegetables.

Furthermore, developing strategies that address constructs within social cognitive theory have been shown to increase the likelihood of influencing behavior changes [35]. To accomplish this, the content of our dietary counseling intervention was based on social cognitive theory, which stipulates that behavior is determined by reciprocal interaction of personal cognitive factors, socio-environmental factors and behavioral factors. Within these constructs, we focused on, knowledge [36], self-efficacy [37], intentions (i.e., goal-setting) [38,39] and reinforcements [40]. To influence families’ behaviors related to salt we chose a pragmatic approach consisting of a 14-week program. It included a 2 h group introduction, followed by a 1 h family meeting and two follow-up telephone interviews with one of the parents.

In planning the workshops for the group introduction, it was important that both children and parents got involved. At the nutrition facts label workshop, participants learned to read labels, to choose foods with less salt. There were examples of keyhole labeled products (easy for children to identify) and nutrition facts labels showing the more processed the higher the salt content of the food. Families received the Heart Foundations shopping guide, size of a credit card, indicating max levels of salt/100 g in usual foods [41]. In the flavor workshop, activities were planned to work around all the aspects of taste, including, smell, texture, mouthfeel, vision. E.g., lemon was tasted to identify sour taste buds and raw and baked carrots were compared for sweetness. The tapas bar served toasted almonds (with no salt) instead of olives, tomato bruschetta to give umami, cottage cheese and boiled potatoes as an alternative to the standard salty ham and cheese. Families were able to spice their own tapas with fresh herbs and spices.

At the individual meeting participants’ motivation was measured on a scale from zero (no motivation) to ten (highly motivated). This was used to ask families, what could motivate them further [42]. To support choice of foods low in sodium a “traffic-light list of foods” was developed: Eat all from the green list (0–0.8 Na g/100 g), less from the yellow and very little from the red (>2.5 g Na/100 g).

Motivation e-mails were sent on a weekly basis (13 mails in total) with inspiration on how to change salt lowering messages into praxis. Furthermore, they build on the advice given at the individual consultations and included taste-good-with-less-salt recipes for breakfasts, sandwich spreads, dinners, sauces and snacks as well as shopping guides to low salt alternatives, hyperlinks to videos on children talking about taste and games to print out and play.

#### 2.10.1. Salt Intake Screener

A salt intake screener was filled out by participants of the dietary counseling group to evaluate their dietary salt habits. The screener was developed for the dietary counseling group to provide a simple and short self-administered questionnaire to detect the participants’ habitual salt consumption and identify areas for potential changes to reduce actual salt intake.

The salt intake screener included a total of 14 questions. Questions were based on a literature review, knowledge on sources of sodium in the Danish diet as well as opinions from project partners. Before the trial the salt screener was checked for face validity by food and nutrition experts and the salt screener scores will be evaluated against participants’ 24 h urine collection results in STRIVE.

The construction of the salt intake screener was mainly based on eating occasions (i.e., breakfast, lunch, dinner and snacks). It was assumed that the best translation of dietary recommendations to practice requires identifying the actual foods consumed that make up the meals [43,44].

The eating occasions contributing the greatest amount of sodium on the day has been found to be dinner and lunch followed by snacks and breakfast [45]. Consequently, the questionnaire included one question related to breakfast intake, five questions related to lunch intake, five questions related to dinner intake and one question related to snack intake. Further, one question was included on daily fruit and vegetable intake, as public health messages that emphasize diet patterns high in vegetables and fruits may aid to reduction in dietary sodium intake as well as increasing the dietary potassium intake [45]. In line with this, Rasmussen et al., found salt content in salad to be significantly lower than the salt content in other meal components [46]. Finally, one question related to purchasing practices of bread, was included in the salt intake screener, as strategies to support reduction in salt intake include reading labels to choose lower sodium options.

#### 2.10.2. Standardization of Dietary Counseling

To standardize the dietary consultations a manual with forms for each step of the dietary counselling was developed. The nutritionists trained each other before the first consultation. Experiences and counseling were shared between nutritionists to ensure coherence in advice giving. Each consultation with the families was reported in a shared logbook.

### 2.11. Outcome Measures

After signing the informed consent form, participants had their health examined at baseline and at the four months follow-up. Participants collected three 24 h urine collections and registered their dietary intake for seven days after the health examination. The intervention ended after the final urine collections and dietary records were completed. All outcomes are presented in Table 3 and described below.

#### 2.11.1. Primary Outcome Measure

The primary outcome is change in sodium intake measured by three consecutive twenty-four-hour urine collections (one in children) at baseline and four months follow-up. Although burdensome, the 24 h urinary collection method was chosen, because salt excretion as a measure of salt intake is more accurate than dietary assessment methods, which often underestimate dietary sodium intake due to underreporting and difficulties quantifying sodium concentration in recipes, and discretionary salt.

##### Urine Collection

The urine collection was carried out simultaneous with a 7-day dietary record. Participants were instructed to follow their usual routines during the collection.

Participants received a brown bottle (3 L), a smaller ‘visiting bottle’ (0.5 L), a large bottle (5 L), a funnel and a urinary hat for the toilet, to aid urine collection. Urine monovettes (Sarstedt, Nümbrecht-Rommelsdorf, Germany) for collection of urine aliquots after the completion of each 24 h collection period. A pen to mark containers and monovettes with name, day and volume. For validation adults also received 3 × 3 80 mg *para*-aminobenzoic acid (PABA) tablets (Glostrup Hospital Pharmacy). A sheet to register beginning and ending of the collection periods, PABA administration and exceptions to the protocol (i.e., estimation of urine loss, medicine).

Adults were informed to collect 24 h urine for three consecutive days (1 weekend day, 2 working days) and children for 1 day (weekend day). All participants received verbal and printed instructions (including a video link) on how to collect 24 h urine: All urine had to be collected during a 24 h period starting from the second urine sample on the morning of the collection day and ending with the first urine sample from the following morning. The morning, after completion of the 24 h urine collection participants marked the volume and day of the collection on the container and registered values in the data sheet. Also, the time of start and finish of the urine collections, and the time of taking the PABA tablets were recorded together with deviations to the instructions. After volume recording, the urine in the container was mixed before taking out aliquots. Hereafter monovettes were frozen at home until returned to CCRP. Containers were rinsed with water and adults could resume their next 24 h urine collection.

A well-trained health worker checked the readings of the total volume marked on the containers and urine aliquots were stored at −20 °C before being transported to the certified laboratory.

Aliquots were further stored at −80 °C for four months before sodium, potassium and creatinine analysis and up to one year for PABA analyses.

Urinary sodium and potassium were measured by potentiometric and creatinine by colorimetric slide tests (Vitros 5.1 FS/5600 Ortho Clinical Diagnostics, Raritan, NJ, USA).

The HPLC method was applied for the determination of PABA [47]. Based on the participants daily recordings of diuresis the 24 h-values of sodium, potassium and creatinine were determined. Mean values were calculated in adults, while children only had one single measurement. In adults with less than three urine collections, it is mentioned, how many days the mean was based on.

PABA is an accepted objective marker to verify completeness of 24 h urine sampling in adults [48,49,50]. The underlying assumption is that PABA is excreted almost quantitatively in 24 h. On collection days adults ingested 240 mg of PABA, divided into three doses of 80 mg (one with each main meal). According to the HPLC method applied a PABA recovery in the urine above 77.9% of total ingested dose indicates urine has been collected for 24 h [47]. However, PABA recovery levels above 105% was regarded as mistaken. If PABA recovery was not available urine collections with collection time less than 22.5 h or more than 25.5 h were excluded as well as urine collections with volume <500 mL/24 h for adults and <300 mL/24 h for children [6].

#### 2.11.2. Secondary Outcome Measures

Secondary outcomes are change in potassium and sodium to potassium ratio measured by the 24 h urine collections, change in blood pressure, blood lipids (triglyceride, total cholesterol, HDL, LDL), hormones (renin, aldosterone, methoxycatecholamines), salt sensitivity and preference and overall dietary intake.

##### Assessment of Blood Pressure and Biochemistry

All participants had their BP measured thrice with an electronic blood pressure monitor (Microlife; Widnau, Switzerland) and fitting cuffs after five minutes of rest in sitting position. Systolic and diastolic BP were calculated as the average of the last two measurements [51].

Blood tests were optional in children 10–17 years old and mandatory in adults. Blood was not drawn from younger children. All adults and children older than 10 years were asked to be fasting minimum two hours before arrival at CCRP. Blood samples were taken after 30 min at complete rest in supine position. Blood samples were transferred to the laboratory (Dept. of Clinical Biochemistry, Glostrup Hospital) for analysis within the first two hours after collection and otherwise centrifuged and stored at 5 °C until the day after or frozen at −80 °C for later analysis.

Total cholesterol, triglyceride and high-density lipoprotein cholesterol (HDL) were sampled in heparin glasses and centrifuged, whereby content was determined in plasma with colorimetric slide test (Vitros 5.1, Ortho Clinical Diagnostics, Raritan, USA). Very-low--density lipoprotein cholesterol (VLDL) and low-density lipoprotein cholesterol (LDL) were calculated from VLDL = 0.45 * triglyceride; LDL = Total Cholesterol–HDL-VLDL. Blood samples for glucose measurement were taken in citrate buffer-fluoride mixture (FC-Mixture) glasses and analyzed by a colorimetric slide test. Glycated haemoglobin (HbA1C) was sampled in EDTA glasses and measured by HPLC. Aldosterone and renin were sampled in EDTA glasses, centrifuged and plasma stored for 4–8 months at −80 °C before analysis by chemiluminescent immunoassay. Epinephrine and norepinephrine were sampled in EDTA glasses and cooled on ice immediately, then centrifuged and plasma stored 4–8 months at −80 °C. The methoxycatecholamines-metanephrine and normetanephrine—are metabolites of epinephrine and norepinephrine, respectively, and were quantified by solid phase extraction followed by liquid chromatography mass spectrometry using a Thermo Q Exactive Plus (Thermo Scientific, Waltham, MA, USA).

As part of the health examination a spot urine sample was taken and send to the laboratory for determination of sodium, potassium, and creatinine content. Albumin content was estimated by micral test (Roche, Mannheim, Germany) to check that inclusion criteria of albumin ≤300 mg were fulfilled.

##### Salt Sensitivity and Preference

Due to the complexity of the salt sensitivity and preference tests children younger than 10 years did not participate in these tests. Salt sensitivity and preference tests were completed in fasting adults and children >10 years. To ensure the same wording, instructions were read by a project worker from a card in the beginning of each test.

Salt taste detection-and recognition thresholds were measured by a staircase procedure with eight different concentrations of salt as described by International Standard ISO 3972 [52]. The concentrations were graded 0.16, 0.24, 0.34, 0.48, 0.69, 0.98, 1.40 and 2.00 g/L. Salt taste detection was measured as the lowest concentration of salt in water which was distinguished from water irrespective of its taste. Recognition threshold was measured as the lowest concentration of salt in water which was correctly identified as salt taste. Because participants were aware of the study being related to salt, it was speculated that some might register their recognition threshold already at the detection threshold. To avoid this, sweet taste detection and recognition threshold were included in the test, with concentrations of 0.34, 0.55, 0.94, 1.56, 2.59, 4.32, 7.20 and 12.00 g/L. The order in which participants received the two staircase procedures was random.

To evaluate salt taste preferences three breads with different salt content (0.4, 0.8 and 1.2 g/100 g) were evaluated using a 7-point hedonic scale. Participants received three pieces of each bread to compare and water to clean the mouth between the tastings. The order of the bread pieces tested was random.

##### Dietary Assessment

Participants (parents assisted their children) recorded their dietary intake using a web-based dietary assessment software for seven consecutive days [53]. The web based dietary assessment software was originally developed and validated for children aged 8–11 years and slightly customized to fit an adult study population [53,54]. At least one weekend day and three week days of food reporting had to be completed for inclusion of the study participant in the analysis [55]. The dietary assessment software is structured according to a typical Danish meal pattern covering breakfast, lunch, dinner and three in-between meals (morning, afternoon and evening). The participants estimated the amount consumed by selecting the closest portion size among four different digital images in eighty photograph series. Internal checks for frequently forgotten foods (spreads, sugar, sauces, dressings, snacks, candy and beverages) were included. Furthermore, the participants reported the intake of nutritional supplements and whether the day represented a usual or unusual intake, including reasons for unusual intakes such as illness or birthday party. If a participant failed to report for a day, the participant was reminded by e-mail the following day, and a text message the second day, if they still missed to report the intake [53].

Intakes of food items, energy and nutrients were calculated for each study participant as an average intake per day using the software system General Intake Estimation System (GIES) v1.000.i6 (National Food Institute, Technical University of Denmark, Kgs. Lyngby, Denmark) and the Danish Food Composition Databank version 7.0 (National Food Institute Technical University of Denmark, 2009). Recipes were updated to better reflect potassium and sodium content.

#### 2.11.3. Additional Variables

##### Anthropometry

As part of the health examination height was measured without shoes to the nearest cm, weight without shoes and overcoat to the nearest kg and body mass index (BMI) was calculated (kg/m^2^). Waist and hip circumferences were measured in cm and waist/hip ratio calculated. Fat percentage was determined from impedance.

##### Covariates

Descriptive variables and possible confounders and modifiers for explorative analyses were collected. At baseline participants answered a questionnaire on gender, age, marital status, occupation, education, health (diagnosed with cancer, diabetes, hypertension, high cholesterol, myocardial infarction, stroke or coeliac disease) and life style (physical activity, smoking, alcohol and dietary habits). At follow-up participants answered additional questions on physical activity, food choice and satisfaction with the bread. Overall the questions were developed and applied for use in Danish National Surveys [56], however, questions on bread satisfaction were developed for this study.

### 2.12. Data Management

The data are handled in accordance with the rules from the Danish Data Protection Agency. Health data was entered directly into a data application developed for the study. The data application helped identifying erratic entries at the examination. Paper forms were entered by double entry. Data entry was validated by checks for valid values and range checks. Data has been exported to SAS Enterprise Guide 9.4 (SAS institute, Inc. Cary, NC, USA). The electronic database and data are stored on a secure computer server with personal login access authorized by the primary investigator. The primary investigator has access to the full data set (blinded to group allocation) and co-investigators will be given access when needed.

### 2.13. Statistical Analysis Plan

#### 2.13.1. Recruitment and Withdrawals

Recruitment rates and numbers of withdrawals and dropouts were reported along with reasons for exclusions, dropouts and withdrawals.

#### 2.13.2. Baseline Data

Descriptive data will be presented as means (std) or medians (with range) for continuous data depending on the distribution of the variable. Categorical variables will be presented as frequencies and percentages.

#### 2.13.3. Compliance

The family is regarded as compliant, with the salt intervention, when at least 80% of the bread had been collected. Compliance sheets were filled in by families on a weekly basis to document whether all participants had eaten the intervention bread. A participant was defined as a compliant bread consumer, when reporting to eat the intervention bread at least 80% of the days in the returned documents.

Completeness of urine collection was evaluated by PABA (incomplete: <77.9%; mistaken >105%). When no PABA results were available completeness was evaluated by collection time (incomplete <22.5 h or >25.5 h) and volume (incomplete: <300 mL in children; <500 mL in adults).

Based on Goldberg’s cut-off method [57] updated by Black [58] prevalence of under-reporting and over-reporting of dietary intakes will be identified and days with unrealistic energy intakes will be excluded accordingly.

Families were compliant to the dietary counselling intervention, when at least the individual meeting and one follow-up call had been completed.

#### 2.13.4. Primary Analyses of the Primary Outcome

The purpose of the primary analysis is to test the trial hypothesis that food reformulation alone or in combination with dietary counseling is an effective strategy to reduce salt intake. Change in participants mean sodium excretion will be determined to compare treatment effects between each of the treatment groups with the control group as well as in between the two treatment groups. All three comparisons will be made using the same statistical model. Estimation of treatment effect for the primary outcome will be calculated using an “intention to treat” analysis, including all participants regardless of the adherence to the intervention and dropouts. To create a full analysis data set, missing data for the primary outcome will be imputed using multiple imputation with 100 samples and including cluster effects if possible, regarding the small number of subjects per family. To test the effect of the primary outcome we will apply a linear mixed model with endpoint values of sodium excretion as outcome variable, treatment group and baseline sodium excretion as fixed effects, and family ID as random effect. The variance structure will be chosen as compound symmetry unless a structure with different covariance for adult and children have better fit (based on AIC) on a dataset consisting of subject with measurements of the primary outcome at follow-up (before multiple imputation). The estimates or the size of treatment effects will be presented along with 95% confidence intervals and actual *p*-values. Levels of significance will be set at 0.05.

##### Sensitivity Analyses of Primary Outcome

We will perform the following sensitivity analyses:

(1) Complete case analyses

(2) Per protocol analysis; analyses will be carried out for compliant bread consumption and complete urine collection. Participants with excluded urine collections will remain in the analyses if they have at least one valid urine sample at both time points.

(3) Adjustment for group imbalances: To examine the influence of potential confounders linear mixed model analyses will be conducted. A priori age, sex, BMI will be adjusted for. If relevant and differences between treatment groups are larger than 5%-point adjustment for education, physical activity, smoking, and alcohol will also be included in the model. Multiple imputation will be used to impute missing values for these variables.

##### Subgroup Analyses of Primary Outcome

Results will be presented for all participants and for a subgroup of children <18 years and a subgroup of adults ≥18.

#### 2.13.5. Analyses of Secondary Outcomes

The purposes of the analyses of secondary outcomes are threefold:(1)To examine the effects of different levels of salt reduction on selected cardiovascular risk factors:(a)Potassium and sodium to potassium ratio (measured in 24 urine)(b)Blood pressure, blood lipids (triglyceride, total cholesterol, HDL and LDL)(c)The renin-angiotensin system (renin, aldosterone measured in blood)(d)The sympathetic nervous response (metanefrines measured in blood)(2)To test effects of different levels of salt reduction on salt taste sensitivity and preference(3)To test effects of different levels of salt reduction on the overall dietary intake and identify possible explanations of salt intake and salt excretion levels.

For each of the secondary outcomes an intention to treat analysis will be performed with multiple imputations. As described for the primary outcome, we will apply linear mixed models with endpoint value as outcome, treatment group and baseline value variable as fixed effects, and family ID as random effect.

Variables will be tested for distribution. All skewed variables will be transformed by relevant transformation before the analyses. Experience from other literature suggests log transformation of triglyceride, renin and aldosterone.

A priori age, sex and BMI will be adjusted for (smoking will also be adjusted for in analyses of salt taste sensitivity and preference). If relevant and differences between treatment groups are larger than 5%-point adjustment for education, smoking, physical activity and alcohol will be included in the model as well. Multiple imputation will be used to impute missing values for these variables.

Results will be presented for all participants and for a subgroup of children <18 years and a subgroup of adults ≥18. However, Blood tests will not be drawn from all children, why analyses of blood lipids, hormones and metanefrines will only include participants ≥ 18 years. Salt sensitivity and preference will be analysed on a subgroup of children 10–15 years and for participants >15 years.

To investigate the effect on cardiovascular risk factors and on salt taste and sensitivity further analyses will be done for participants in the intervention groups that achieved a moderate (1.25–3.0 g/d) and a high (>3.0 g/d) reduction in salt intake during the four months intervention. In addition to the analyses described the salt intake screener will be evaluated. All analyses will be carried out using SAS 9.4 (SAS Institute Inc.) or R version 3.5.3 ( Core Team (2019), Vienna, Austria).

## 3. Discussion

The aim of the STRIVE intervention study was to explore the feasibility and health effects of different strategies to gradually reduce salt intake in families. This paper describes STRIVE with regards to: rationale, study design, sample eligibility criteria and recruitment, intervention content, examinations, data management and planned analyses.

Bread was chosen for the reformulation strategy, because bread is the biggest contributor of salt in the Danish diet and the salt content of the bread can be blinded to the participants. However, national strategies to reduce population salt intake cannot depend on a single food alone, other processed foods must also be reformulated in terms of salt content. Furthermore, dietary changes are needed to obtain the recommended daily level of salt intake, why a dietary counseling strategy was developed too.

To avoid data-driven analyses and ensure robust reporting of outcomes the statistical analysis plan for the primary and secondary outcomes of STRIVE was developed before commencing the statistical analyses. Analyses of STRIVE will test the cardiovascular consequences of the applied strategies to reduce salt intake in families, including potential adverse effects.

Procedures have been developed to minimize lack of compliance and incomplete data collection in the study, however, it cannot be completely avoided. Although, the repeated 24 h urine collections and 7-day dietary records are the most valid methods to estimate salt intake and the overall dietary intake on an individual level, they are known to be burdensome and limited by under-collection/under-reporting. These limitations will be adjusted for by applying imputations of missing values in the intention to treat analysis and performing per protocol and sensitivity analyses of complete data. The burdensome data collection methods may result in selection bias as mainly highly motivated families will enroll. Yet, free bread from a quality baker may result in a more representative participation. Furthermore, a gradual reduction of salt content is likely to have the same physiological response in healthy individuals regardless of background.

Randomization was based on families and not on participants. Effects of clustering were accounted for in the power calculation, the imputations and by including family ID as random effect in the linear mixed models.

## 4. Conclusions

STRIVE is developed to ensure that the results easily can be transferred to the ongoing prevention efforts, both for children and adults. Providing families with reformulated bread in terms of reduced salt content is a new way to investigate the effect of salt reduction. The strategy is simple and in accordance with the official recommendation of a gradual reduced salt content in processed food. The results will qualify mechanisms affected during a gradual reduction in salt intake for a longer period including the effect on the renin-angiotensin-aldosterone system, sympathetic nervous system and development in salt preferences.

New educational tools to measure and promote changes in salt intake in families were developed for the dietary counseling strategy. It is our hope that the methods we have outlined may prove useful to other researchers planning similar trials.

## Figures and Tables

**Figure 1 ijerph-16-03532-f001:**
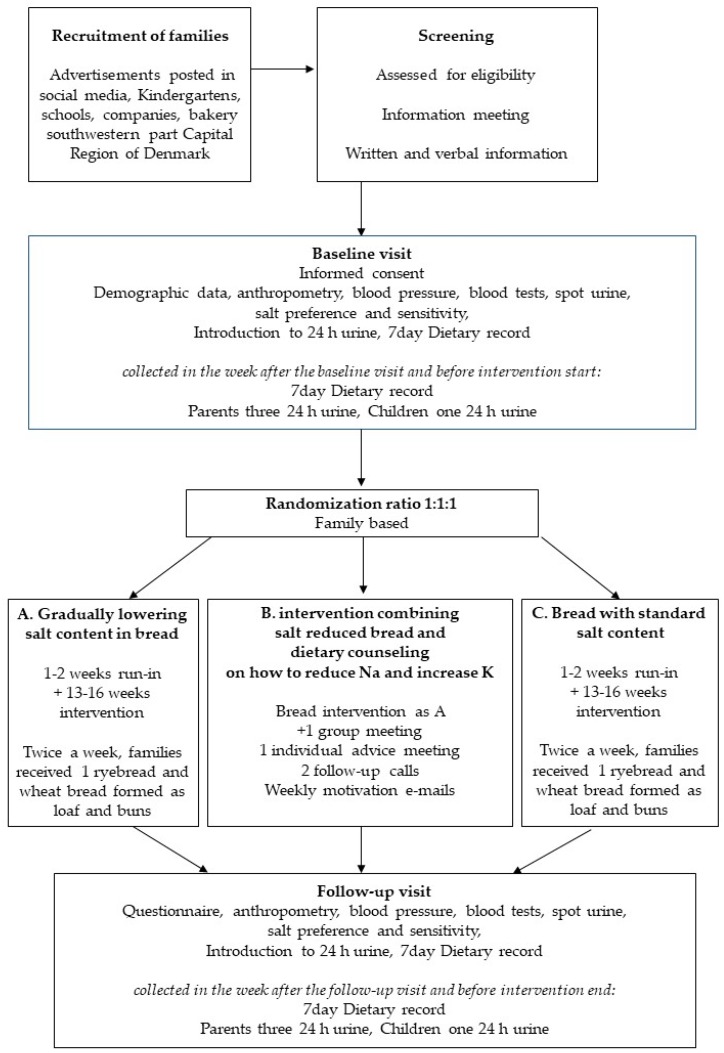
Trial flow diagram. Participants were recruited as families, screened for eligibility and informed about the trial. After informed consent the baseline examination was completed followed by allocation to trial group. Then dietary record and urine was collected for one week, while consuming own bread before intervention start. Depending on when collection of dietary data and 24 h urine ended the run-in period with standard bread varied from one to two weeks. Participants consumed intervention bread the following 13–16 weeks, depending on the date of the follow-up visit. The follow-up visit included one week of 7-day dietary record + three 24 h urine collections while consuming intervention bread. Hereafter, the intervention ended.

**Table 1 ijerph-16-03532-t001:** Salt content in bread—Six weeks of gradual reduction, hereafter same level as week six.

Week from Intervention Start	Bread, Reduced Salt Content		Bread, Standard Salt Content	
	Rye Salt (g/100 g)	Wheat Salt (g/100 g)	Rye Salt (g/100 g)	Wheat Salt (g/100 g)
1	1.2	1.2	1.2	1.2
2	1.2	1.2	1.2	1.2
3	1.0	1.0	1.2	1.2
4	0.8	0.8	1.2	1.2
5	0.6	0.6	1.2	1.2
6	0.6	0.4	1.2	1.2

**Table 2 ijerph-16-03532-t002:** Application of theory on salt reduction to dietary counseling.

Activity	Social Cognitive Theory	Advice and Skill Development	Tools	Measurements
General introduction	Knowledge	Salt and Health	Shopping guide	Salt intake screener
		How to reduce salt and increase potassium	Keyhole label	
		Nutrition facts labels	Workshops	
		More taste with less salt		
Individual advice	Self-efficacy	Rethink food choice alternatives with less salt	Traffic light: foods with lower salt content	Motivation, Barriers, Focus points/goal setting
		Promote keyhole, nutrition declarations		
Two follow-up calls	Re-enforcement	Encouragement		Follow-up calls
		Guidance		Motivation, Barriers, focus points
Weekly motivation e-mails	Re-enforcement	Information, recipes		

**Table 3 ijerph-16-03532-t003:** Outcome variables and assessment tools.

Assessment	Outcome	Method
Demographics	Gender, age, family, marital status, education, work	Questionnaire
Lifestyle	Physical activity, smoking, alcohol, salt and bread consumption, food habits	Questionnaire
Medical history	Hypertension, heart disease, cancer, diabetes, cholesterol, coeliac disease	Questionnaire, Interview
Anthropometry	Height, weight, BMI, arm, hip and waist circumferences, fat percentage	Standardized measurements by nurse, impedance
Blood pressure	Pulse, diastolic Blood Pressure (BP) Systolic BP	Three resting BP’s, 5 min rest–average of last 2 BP’s
Biochemistry	P-glucose, glycated haemoglobin (HbA1C), Triglyceride, Cholesterol, high-density lipoproteins (HDL), low-density lipoproteins (LDL),	Semi fasting (min 2 h) blood samples after 30 min rest
	renin, aldosterone, catecholamine metabolites	24 h urine
	U-albumin, U-sodium, U-potassium, U Creatinine	
	U-albumin	Spot urine, micral test
Salt sensitivity/preference	Salt taste detection- and recognition thresholds staircase procedure: eight salt concentrations	Taste of salt solutions
	Preference bread 0.4/0.8/1.2 g salt/ 100 g	Taste of bread
24 h urine collection	24 h-sodium, 24 h-potassium, 24 h-Creatinine, *para*-aminobenzoic acid (PABA)	Adults: 3 days 24 h urine + PABA
		Children: 1 day 24 h urine
7-day dietary assessment	Energy, macro- and micronutrients, foods, meal pattern, eating at home/out	Validated web-based dietary record, estimated portion sizes from photos
Bread intervention	Bread collection twice a week registration	Four months of bread intervention:
	Weekly test of salt content in bread	Intro week 1–2 after completion baseline dietary record and urine collection
		Six weeks gradual reduction in salt
	Participants evaluation of bread consumed	*End* after completion follow-up dietary record and urine collection
Dietary counseling	Salt intake, potassium	Salt intake screener, interview, follow-up telephone interviews
	Motivation, barriers, focus points	
	Actions applied to reduce salt intake	Evaluation questionnaire

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
