# Peer review of "Salt Reduction Intervention in Families Investigating Metabolic, Behavioral and Health Effects of Targeted Intake Reductions: Study Protocol for a Four Months Three-Armed, Randomized, Controlled “Real-Life” Trial"

_ijerph, 2019, doi:10.3390/ijerph16193532_

Round 1

Reviewer 1 Report

The authors describe an important large-scale intervention to reduce salt intake in Denmark residents. The protocol is described thoroughly and methods are supported with appropriate references. A strength is that families with children are included.

Comments and suggestions:

Title: Shorten. "Salt" needs only one mention.

Abstract

Line 19: remove first use of "population."

21, 32: Longer than what?

23: "salt-reduced"

26: Suggest changing "A+B" to "B"

28: specify what is measured in urine

32: Suggest change to "in compliance with" vs. "analogously"

Introduction

50: Change "Much less studies in this area are done among children" to "Fewer studies in this area have been conducted among children"

Materials and Methods

Figure 1: Add time to the figure, especially to the intervention (# weeks) and time until follow-up and final visit.

213, 288: the salt screener should be validated prior to use in the trial

268: keyhole labels are not broadly known. Provide a reference or describe briefly.

273: taste buds

415: is water "softened" (Mg, Ca removed and Na, K added to household water) in Denmark? Consumption of softened water would be important to measure.

476: How did you deal with food records showing severe under- or over-reporting? Did a dietitian ever contact a participant when clarification was needed?

516-19: Here and throughout, specify measures as urine or blood

Author Response

Thank you for good comments. I have the following response to reviewer 1: 

Point 1:

Title: Shorten. "Salt" needs only one mention.

Response 1: Deleted “in salt” in title

Point 2: Abstract

Line 19: remove first use of "population."

21, 32: Longer than what?

23: "salt-reduced"

26: Suggest changing "A+B" to "B"

28: specify what is measured in urine

32: Suggest change to "in compliance with" vs. "analogously"

Response 2:

Deleted first use of populaton

21: Changed text to lasting at least four weeks are required

32: Deleted longer period

23: salt reduced changed to salt-reduced

26: Changed "A+B" to "B" line 26, 99, 150, 207, figure 1,

28: Changed sentence to Secondary outcomes are change in urine measures of potassium and sodium/ potassium ratio,

32: Changed "analogously" to "in compliance with".

Point 3:

Introduction

50: Changed "Much less studies in this area are done among children" to "Fewer studies in this area have been conducted among children"

Response 3:

49: Changed "Much less studies in this area are done among children" to "Fewer studies in this area have been conducted among children"

Point 4:

Materials and Methods

Figure 1: Add time to the figure, especially to the intervention (# weeks) and time until follow-up and final visit.

Response 4:

Figure 1: Time (weeks) added to the figure and figure text, regarding intervention and time until follow-up and final visit.

Point 5:

213, 288: the salt screener should be validated prior to use in the trial

Response 5: Formulation in line 297 has been reformulated and a line has been added in line 299 Before the trial the salt screener was checked for face validity by food and nutrition experts. and the salt screener scores will be evaluated against participants’ 24 hr urine collection results in STRIVE.

Point 6:

268: keyhole labels are not broadly known. Provide a reference or describe briefly.

Response 6 The Keyhole label was introduced in line 235, and I have added extra information about the label here: Before intervention start, the bakery gradually reduced the salt content of their rye bread from about 1.7 g/100 g to meet the keyhole label maximum salt content of 1.2 g/ 100 g. The keyhole label is a Nordic label helping consumer to choose healthier foods. Foods labelled with the Keyhole symbol contain less fat, sugars and salt and more dietary fibre than food products of the same type not carrying the symbol [32].

Point 7:

273: taste buds

Response 7: corrected spelling to taste buds

Point 8:

415: is water "softened" (Mg, Ca removed and Na, K added to household water) in Denmark? Consumption of softened water would be important to measure.

Response 8: No, water in Denmark is not softened. We do not have tradition for measuring it in our dietary surveys. I have not added information about this to the paper.

Point 9:

476: How did you deal with food records showing severe under- or over-reporting? Did a dietitian ever contact a participant when clarification was needed?

Response 9:

We were not able to evaluate dietary records at the same time as the data collection, why dietitians did not clarify unrealistic food records during the trial. However, data are collected in order to be able to evaluate prevalence of severe under or over reporting according to Goldenberg and Black. In addition to energy intake dietary records will be evaluated according to unrealistic food recording pattern / nutrient intake.

The following lines have been added to line 481: Based on Goldberg’s cut-off method updated by Black [57,58] prevalence of under-reporting and over-reporting of dietary intakes will be identified and days with unrealistic energy intakes will be excluded accordingly.

Refences have been added:

Goldberg G, Black A, Jebb S, Cole T, Murgatroyd P, Coward W, Prentice A: Critical evaluation of energy intake data using fundamental principles of energy physiology: 1. Derivation of cut-off limits to identify under-recording. Eur J Clin Nutr 1991, 45(12):569-581.

Black AE: Critical evaluation of energy intake using the Goldberg cut-off for energy intake: basal metabolic rate. A practical guide to its calculation, use and limitations. Int J Obes Relat Metab Disord 2000, 24(9).

Point 10:

516-19: Here and throughout, specify measures as urine or blood

Response 10: measures as urine or blood has been specified

Reviewer 2 Report

This manuscript described the protocol for the salt reduction intervention in Denmark. It is a real-life trial to test whether salt reduction of bread alone or in combination with dietary counselling are effective strategies to reduce salt intake in families. It is well-designed and described in detail. I expect the results. I have some comments for consideration.

Selection bias: While the design is scientific and proper, on the other hand, it might be somewhat burdensome for the participants. Therefore I am afraid only the highly motivated families are willing to enter the study. In that case, the results cannot always be applicable to general population in Denmark. It may be hard to collect 24 hr urine in consecutive 2 working days. It may be hard to collect 24 hr urine in 3-5 (9) year old children.

Minor

Line 48, it may be better to use EUR or USD instead of DKK. Line 205, it should be 67% instead of 60%

Author Response

Response to Reviewer 2 Comments:

Thank you for good points. Please, find response below:

Point 1:

Selection bias: While the design is scientific and proper, on the other hand, it might be somewhat burdensome for the participants. Therefore, I am afraid only the highly motivated families are willing to enter the study. In that case, the results cannot always be applicable to general population in Denmark. It may be hard to collect 24 hr urine in consecutive 2 working days. It may be hard to collect 24 hr urine in 3-5 (9) year old children.

Response 1:

We agree that selection bias is likely, due to the burdensome data collection methods and only the highly motivated participants may sign up and complete the study.

Motivation is likely to be influenced by education and socio-economic background of the participants, why SES will be described, in papers analysing the intervention results.

The following text has been added to the discussion – Line 572: The burdensome data collection methods may result in selection bias as mainly highly motivated families will enroll. Yet, free bread from a quality baker may result in a more representative participation. Furthermore, a gradual reduction of salt content is likely to have the same physiological response in healthy individuals regardless of background.

Point 2:

Line 48, it may be better to use EUR or USD instead of DKK.

Response 2: Changed to 296 million USD

Point 3:

Line 205, it should be 67% instead of 60%

Response 3: Correct, it is 67%
